# Neuroprotective Effects of a Novel Demeclocycline Derivative Lacking Antibiotic Activity: From a Hit to a Promising Lead Compound

**DOI:** 10.3390/cells11172759

**Published:** 2022-09-04

**Authors:** Rodrigo Tomas-Grau, Florencia González-Lizárraga, Diego Ploper, César L. Avila, Sergio B. Socías, Pierre Besnault, Aurore Tourville, Rosa M. Mella, Patricia Villacé, Clarisa Salado, Clémence Rose, Blandine Seon-Méniel, Jean-Michel Brunel, Laurent Ferrié, Rita Raisman-Vozari, Patrick P. Michel, Bruno Figadère, Rosana Chehín

**Affiliations:** 1Instituto de Investigación en Medicina Molecular y Celular Aplicada (IMMCA) (CONICET-UNT-SIPROSA), Pasaje Dorrego 1080, San Miguel de Tucumán 4000, Argentina; 2Paris Brain Institute-ICM, Inserm, CNRS, Sorbonne Université APHP, Hôpital de la Pitié la Pitié-Salpêtrière, 75013 Paris, France; 3Innoprot SL, Parque Tecnológico de Bizkaia, Edificio 502, 48160 Derio, Spain; 4BioCIS, Université Paris-Saclay, CNRS, 92290 Châtenay-Malabry, France; 5UMR_MD1 “Membranes et Cibles Thérapeutiques”, U1261 INSERM, Aix-Marseille Université, 13385 Marseille, France

**Keywords:** Parkinson’s disease, novel tetracycline, neuroprotection

## Abstract

The antibiotic tetracycline demeclocycline (DMC) was recently reported to rescue α-synuclein (α-Syn) fibril-induced pathology. However, the antimicrobial activity of DMC precludes its potential use in long-term neuroprotective treatments. Here, we synthesized a doubly reduced DMC (DDMC) derivative with residual antibiotic activity and improved neuroprotective effects. The molecule was obtained by removal the dimethylamino substituent at position 4 and the reduction of the hydroxyl group at position 12a on ring A of DMC. The modifications strongly diminished its antibiotic activity against Gram-positive and Gram-negative bacteria. Moreover, this compound preserved the low toxicity of DMC in dopaminergic cell lines while improving its ability to interfere with α-Syn amyloid-like aggregation, showing the highest effectiveness of all tetracyclines tested. Likewise, DDMC demonstrated the ability to reduce seeding induced by the exogenous addition of α-Syn preformed fibrils (α-Syn_PFF_) in biophysical assays and in a SH-SY5Y-α-Syn-tRFP cell model. In addition, DDMC rendered α-Syn_PFF_ less inflammogenic. Our results suggest that DDMC may be a promising drug candidate for hit-to-lead development and preclinical studies in Parkinson’s disease and other synucleinopathies.

## 1. Introduction

In recent decades, tetracyclines (TCs) have been considered beyond their antibiotic effects. In fact, they have anti-inflammatory and anti-apoptotic activities and can exert a variety of biological actions, including the inhibition of proteolysis, angiogenesis, and tumor metastasis [1,2]. The pioneering work of Yrjanheikki and colleagues published in 1998 demonstrated that minocycline was neuroprotective in an animal model of ischemia [3]; moreover, it was recently reported that TC use in rosacea-treated patients was associated with a reduction in the risk of developing Parkinson’s disease (PD) [4]. The neuroprotective effects of minocycline have also been described in PD animal models, where they are mainly attributed to the inhibition of microglial activation [5,6]. However, it seems that anti-inflammatory therapy *per se* is insufficient to account for neuroprotection, as PD patients in clinical trials being treated with minocycline showed no improvement [7].

The neuroprotective effect of doxycycline for several neurodegenerative diseases was previously reported [2,8,9]. In our team, we studied the ability of minocycline, doxycycline, and the non-antibiotic CMT-3 to interfere with α-Syn amyloid-like aggregation, since this process is widely considered to be central in triggering and spreading neurodegeneration in PD [10,11]. Our results demonstrated that dimethylamino (DMA) substituent, which are largely involved in antimicrobial activities [12], negatively correlates with anti-aggregant properties of these compounds. In fact, minocycline, which possesses two DMA substituents located at C4 and C7, was incapable of inhibiting α-Syn aggregation and/or disrupting α-Syn preformed fibrils (α-Syn_PFF_). Doxycycline, with only one DMA substituent at C4, interferes with aggregation but does not disassemble α-Syn_PFF_. Finally, CMT-3, without any DMA substituent, is able to block α-Syn amyloid aggregation and even disassemble α-Syn_PFF_ [11]. 

Recently, using a high-throughput drug discovery biosensor platform, demeclocycline (DMC) was selected from a 1280-compound small-molecule library (LOPAC) as a hit compound with enough potential to prevent neurodegeneration induced by the aggregation of α-Syn [13]. Interestingly, among 21 hit compounds identified by the biosensor, DMC showed the highest inhibition of αSyn-induced toxicity in the low nanomolar range (EC50 = 65 nM) and was tenfold more potent than previously described small molecules. Moreover, DMC efficiently rescued primary dopaminergic neurons from α-Syn_PFF_-induced pathology. This data led to positioning DMC as an interesting compound for hit-to-lead optimization. However, the antibiotic activity of DMC strongly decreases its potential for long-term treatments, such as those required for PD and other synucleinopathies. A systematic revision of oral tetracycline administration during 2–18 weeks showed an increase in antibiotic resistance in subgingival, gastrointestinal, and upper respiratory tract flora [14]. It is relevant to consider that the frequency of microbial antibiotic resistance continues to increase and has emerged as one of the most critical problems in medicine.

Here, we aimed to advance the hit DMC compound towards a lead molecule with improved neuroprotective properties and minimal antibiotic activity, thus taking advantage of the neuroprotective potential of DMC while avoiding the antibiotic resistance dilemma. For this, we synthesized a new DMC derivative without DMA substituents at position 4 and reduced the hydroxyl group at position 12a on ring A of the molecule to obtain a novel non-antibiotic TC. The The doubly reduced DMC derivative referred as DDMC, showed only residual antibiotic activity, no toxicity in a dopaminergic cell model, and the ability to significantly diminish α-Syn amyloid-like aggregation at lower concentrations than other previously tested TCs [10,11]. Furthermore, α-Syn_PFF_ in the presence of the novel compound (α-Syn_PFF_+DDMC) resulted in a significant reduction in proinflammatory factors released by primary microglial cell cultures. On the other hand, the fibrils formed in the presence of DDMC (α-Syn_PFF_:DDMC) were less prone to induce aggregation and seeding in SH-SY5Y cells and were less inflammogenic than α-Syn_PFF_. Taken together, these results position DDMC as a novel multimodal drug candidate as the lead molecule of the TC family with promising properties for preclinical and clinical studies in PD and other synucleinopathies.

## 2. Materials and Methods

### 2.1. Synthesis of DDMC

In a 50 mL round-bottom flask, DMC hydrochloride (300 mg, 0.6 mmol, 1.0 eq) was suspended in AcOH (6.2 mL); then, zinc powder (391 mg, 6.02 mmol, 10 eq) was added, and the reaction mixture was stirred for 20 h at room temperature. The resulting solution was filtered through a small pad of Celite with AcOH. The organic phase was extracted with CH_2_Cl_2_, washed with HCl (1 M) and brine, dried over MgSO_4_, filtered off, and concentrated in vacuo. Precipitation in EtOAc/*n*-pentane produced compound 6 as a yellow solid (130.5 mg, 32%). Further purification was performed with preparative HPLC (XSelect 4.6 × 150 mm 5 µm, H_2_O + 0.1%TFA: MeCN, 25:75 to 80:20 over 20 min. tr = 11.8 min).

^1^H NMR (400 MHz, DMSO-*d6*): δ 18.45 (brs, 1H, OH), 14.71 (s, 1H, OH), 11.91 (s, 1H, OH), 9.11 (brs, 1H, NH), 8.72 (s, 1H, NH), 7.59 (d, *J* = 8.9 Hz, 1H), 6.95 (d, *J* = 8.9 Hz, 1H), 5.71–4.88 (brs, 1H, OH), 4.72 (d, *J* = 2.7 Hz, 1H), 3.68 (brs, 1H, CH_12a_), 3.17 (brs, 1H, OH), 2.82 (ddd, *J* = 11.4, 5.6, 2.7 Hz, 1H), 2.46–2.55 (m, 3H), 1.87 (q, *J* = 12.2 Hz, 1H), 1.70 (ddd, *J* = 12.7, 5.6, 2.4 Hz, 1H) ppm. ^13^C{^1^H} NMR (100 MHz, DMSO-*d6*): δ ^13^C NMR (101 MHz, DMSO) δ 195.32, 191.73, 190.84, 177.76, 173.24, 160.12, 140.88, 136.60, 122.16, 118.72, 116.03, 104.33, 99.19, 63.91, 50.96, 37.97, 36.99, 29.05, 26.65 ppm.

HRMS: calculated for C_19_H_17_ClNO_7_ [M+H]^+^: 406.0688, found 406.0694.

### 2.2. Expression and Purification of Human Recombinant α-Syn

Recombinant wild-type human α-Syn was expressed in *Escherichia coli* using the pT7-7 plasmid encoding for the protein sequence. Purification was performed as previously described [15]. Protein purity was assessed using electrophoresis in polyacrylamide gels under denaturing conditions (SDS-PAGE). The stock solution of α-Syn was prepared in 20 mM HEPES, 150 mM NaCl, pH 7.4. Prior to the aggregation assay, the protein stock solutions were centrifuged for 30 min at 12,000× *g* to remove microaggregates. Protein concentration was determined by measuring the absorbance at 280 nm using the extinction coefficient ε_275_ = 5600 cm^−1^ M^−1^.

### 2.3. Antimicrobial Assays

The bacterial strains used in this study were *S. aureus* (ATCC25923), *E. coli* (ATCC25922), and *P. aeruginosa* (PA01). The strains were maintained at −80 °C in 15% (*v*/*v*) glycerol for cryoprotection. Bacteria were routinely grown in Mueller–Hinton (MH) broth at 37 °C.

The susceptibility of bacterial strains to antibiotics and compounds was determined in microplates using the standard broth dilution method in accordance with the recommendations of the Comité de l’Antibiogramme de la Société Française de Microbiologie (CA-SFM) [16]. Briefly, the minimal inhibitory concentrations (MICs) were determined with an inoculum of 10^5^ CFU in 200 µL of MH broth containing two-fold serial dilutions of each drug. The MIC was defined as the lowest concentration of drug that completely inhibited visible growth after incubation for 18 h at 37 °C. To determine all MICs, the measurements were independently repeated in triplicate.

### 2.4. Cell Viability Assay

Human neuroblastoma cell cultures (SH-SY5Y) were grown in Dulbecco’s Modified Eagle Medium (DMEM) supplemented with 10% fetal bovine serum (FBS) and 1% penicillin/streptomycin (PS) at 37 °C and 5% CO_2_. For the cell viability assay, cells were seeded in 96-well plates at 15,000 cells/well and maintained in 100 μL of DMEM supplemented with 2% FBS and 1% PS for 24 h at 37 °C. Afterward, the cells were treated with DMC or DDMC (10 or 50 µM final concentrations in each well). After treatment, the cells were incubated for 24 h at 37 °C and 5% CO_2_. To determine cell viability, a colorimetric MTT metabolic activity assay was used as previously described by Mosmann [17]. All experiments were performed in sextuplicate, and the relative cell viability (%) was expressed as a percentage relative to the untreated control cells.

### 2.5. Protein Aggregation Assays

The aggregation protocol was adapted from previous studies [10]. The different aggregated species were formed by incubating recombinant α-Syn samples (70 μM) in 10 mM PBS, pH 7.4, in a Thermomixer Comfort^®^ (Eppendorf, Germany) at 37 °C under orbital agitation at 600 rpm in the absence or presence of DMC or DDMC.

### 2.6. Thioflavin T Fluorescence Assay

Aggregation studies with α-Syn in the absence or presence of DMC or DDMC were performed by measuring the fluorescence emission of Thioflavin T (ThT) at different time points according to LeVine [18]. Changes in the emission fluorescence spectra were monitored at an excitation wavelength of 450 nm using a Fluoromax-4 spectrofluorometer.

### 2.7. Congo Red Absorbance Spectroscopy

A Congo red (CR) working solution was prepared prior to use by diluting a stock solution to reach a concentration of 10 mM in 10 mM PBS, pH 7.4. A fresh solution was prepared immediately before use. The samples used were obtained after the incubation of 70 µM α-Syn in the presence of DMC or DDMC. For absorbance measurements, the procedure was performed according to Medina et al. [9] with minor adjustments. Each sample was mixed with PBS and CR solution to reach a final dye concentration of 20 µM. After that, a transparent 96-well plate was filled with 200 µL of each prepared sample, vortexed at 23 °C and 300 rpm for 30 min before being read in the same spectrometer. The absorption of each sample was recorded over a range of 400–700 nm using a TECAN Infinite M200 microplate reader. The CR absorbance was calculated according to the equation CRAb = (540Ab/25,295) − (475Ab/46,306), as previously described by Klunk et al. [19]. 

### 2.8. Electron Microscopy 

Samples (50 μL) of a 70 μM α-Syn solution were adsorbed onto glow-discharged 200 mesh carbon film-coated copper grids (Electron Microscopy Sciences) and stained with uranyl acetate (2%). The excess liquid was removed, and the grids were allowed to air dry. The grids prepared in this way were analyzed using Transmission Electron Microscopy (TEM). TEM images were captured using a Philips 301 transmission electron microscope.

### 2.9. In Vitro Protein Seeding Assay

Monomeric α-Syn solutions (70 μM) in 10 mM PBS, pH 7.4, were incubated in a Thermomixer Comfort (Eppendorf) at 37 °C under continuous orbital agitation at 600 rpm to form fibrils. After harvesting, the aggregated species were diluted 1/10 and further incubated with 70 μM of fresh monomers in the absence or presence of 10 μM DDMC. Aliquots were taken from the seeding aggregation reaction at different times and mixed with the ThT probe, according to LeVine [18]. Changes in the emission fluorescence spectra (λ_exc_ = 450 nm) were monitored using a Horiba Fluoromax-4 fluorometer. Non-linear regression was used to fit the kinetic curves to a sigmoidal equation in GraphPad Prism, and the lag phases were calculated using the following equation: T50 − [1/(2 ∗ k)], where T50 is the time at which the fluorescence is halfway between the baseline and plateau values and k is the Hill slope according to Lau et al. [20].

### 2.10. Cell Seeding Assay

Recombinant human α-Syn (70 μM) was solubilized in 10 mM PBS (pH 7.4), filtered, centrifuged, and incubated for 120 h at 37 °C under orbital agitation at 600 rev./min in a Thermomixer comfort (Eppendorf) to form fibrils. SH-SY5Y cells overexpressing the fusion protein α-Syn-tRFP provided by Innoprot (Innoprot #P30707-02) were plated on 48-well plate at a density of 50,000 cells per well and either treated or not treated with 10 μM DDMC for 72 h with a 7 μg/mL suspension of α-Syn_PFF_ aggregates. After treatment, the cells were fixed with 4% (*w*/*v*) paraformaldehyde (PFA) for 20 min at room temperature in the dark, followed by a permeabilization step with 0.1% (*v*/*v*) Triton X-100 in PBS for 20 min at room temperature. Nuclei were finally counterstained with DAPI (1:5000 in PBS) for 5 min and mounted on coverslips for confocal microscopy. Images were acquired with a ZEISS LSM800 confocal microscope at 63× and image analysis was performed using CellProfiler 4.2.1Cambrige, Massachusetts, USA [21]. Cell segmentation was performed with the RunCellPose plugin, while the granularity spectra of the cytoplasm was evaluated with the granularity module.

### 2.11. Immunofluorescence and Lysosome Identification

For studying lysosomes, SH-SY5Y cells were grown to 80% confluency in 24-well plates on round coverslips which were previously washed with ethanol, PBS buffer, and culture medium. After overnight incubation of the cultures with or without DDMC, LysoTracker™ Deep Red (Invitrogen #L12492), diluted 1:1000 in prewarmed medium, was added to SH-SY5Y cells for 5 min at 37 °C in an incubator supplied with 5% CO_2_. Afterward, the cells were washed three times with warm media and fixed with fresh 4% (*w*/*v*) paraformaldehyde (PFA) diluted in PBS for 20 min at room temperature protected from light. After this, the cells were washed for 10 min three times in freshly prepared PBS and mounted for microscopy. For immunostaining of LAMP1-positive vesicles, SH-SY5Y cells were also grown on coverslips in similar conditions to those described before and treated or not treated overnight with DDMC. The next day, the cells were fixed in fresh PFA for 20 min, washed for 10 min three times in freshly prepared PBS, permeabilized by treatment with 0.2% (*v*/*v*) Triton X-100 in PBS for 10 min, and washed again for 10 min three times with PBS. The samples were then blocked with 5% dry milk and 0.5% BSA in PBS (blocking solution) for 2 h. Immunofluorescence was performed with rabbit monoclonal anti-Lysosomal-Associated Membrane Protein 1 (LAMP1) primary antibodies (Cell Signaling #3243S) at a dilution of 1/300 in a solution of 1/10 blocking solution in PBS overnight at 4 °C. The following day, the wells were washed for 10 min three times with PBS, and goat anti-rabbit IgG (H+L) cross-adsorbed to Alexa Fluor 488 secondary antibody (Invitrogen #A11008) at a 1/750 dilution (also diluted in 1:10 blocking solution) and applied for 2 h at room temperature in the dark, followed by a washing step with PBS for 10 min three times. For both, the LysoTracker^TM^ assay and LAMP1 immunostaining, cell nuclei were stained before mounting with DAPI for 5 min and washed three times with PBS. The coverslips were then mounted on microscopy slides with 4 μL of Vectashield Antifade Mounting Media (Vector Laboratories #H1000). The samples were visualized and analyzed with a ZEISS LSM800 confocal microscope at 63x magnification. Image analysis for lysosome quantification was performed with the granularity module in CellProfiler 4.2.1 [21].

### 2.12. Primary Microglial Cell Treatments

Microglial cell isolation: Cultures were generated as described previously [22]. Briefly, polycation coating solutions containing 1 mg/mL polyethyleimine (PEI) were applied to culture vessels for 2 h at 37 °C; then, they were washed with PBS and used for cell seeding. Newborn mice were sacrificed and had their brains rapidly dissected. Two mouse brains were plated onto PEI-coated Corning T-75 flasks (Sigma-Aldrich) with 12 mL of complete medium (DMEM + 10% FBS and antibiotics) and incubated at 37 °C in an atmosphere of 95% air and 5% CO_2_. A total of 10 mL of the culture medium was removed 48 h after plating to eliminate floating debris, and 10 mL of fresh medium was added. No additional medium was added until completion of microglial isolation. To produce subcultures, microglial cells were recovered by trypsin proteolysis using an EDTA (2 mM)-trypsin (0.05%) solution and seeded onto non-coated Nunc 96-well plaques at a density of 20,000 cells per well for 24 h using N5B medium (N5 medium + 5% HS + 0.5% FBS + 5 µM glucose) with 100 µM glycine.

α-Syn preparation for microglial cell treatments: To obtain α-Syn_PFF_ free of endotoxins, we processed monomeric recombinant α-Syn through a high-capacity endotoxin removal spin column (ThermoFisher, #88274) following the manufacturer’s instructions. Then, protein samples were filtered through a 0.21 µm diameter pore membrane and centrifuged for 30 min at 12,000× *g*, and residual endotoxins were quantified using a Limulus Amebocyte Lysate chromogenic endotoxin quantification kit (ThermoFisher, #88288). Monomeric α-Syn stock solutions containing less than 1.2 endotoxin units (EU)/mL were used to obtain α-Syn_PFF_ or α-S_PFF_:DDMC following the same aggregation conditions described above. Before treating microglia cells, the aggregates were sonicated for 10 cycles at high power (30 s on and 30 s off, at 10 °C).

Dialysis: The samples tested in primary cell cultures were dialyzed twice against 10 mM PBS, pH 7.2, for 24 h using Slide-A-Lyzer™ 10 k Dialysis Cassettes (ThermoFisher, Waltham, MA, USA #66380). DDMC removal from samples was confirmed using spectroscopic methods.

Biochemical assays: Tumor necrosis factor (TNF)-α release was measured using an enzyme-linked immunosorbent assay (ELISA) kits from ThermoFisher Scientific (#BMS607-3). ELISA standard curves were generated using a four-parameter logistic curve model (GraphPad Prism 8, GraphPad Software, San Diego, CA, USA). Glutamate (Glut) was measured using the Amplex Red Glutamic Acid/Glutamate Oxidase Kit (#A12221; Invitrogen, Vilebon sur Yvette, France) according to the manufacturer’s instructions. The absorbance of each sample was measured according to the manufacturer’s instructions using a spectrophotometer SpectraMax M4 (Molecular Devices, Sunnyvale, CA, USA).

### 2.13. Statistical Analysis

For the ThT fluorescence assay, Congo red absorbance spectroscopy, and in the cell model protein seeding assay, statistical analyses were performed using one-way ANOVA, *t*-test and Kruskal-Wallis test.

## 3. Results

### 3.1. Non-Antibiotic and Non-Toxic Tetracycline Obtained by Chemical Modifications of DMC

In order to eliminate the characteristic antibiotic activity of TCs, DMC was subjected to chemical modifications. The removal of the DMA substituents at position 4 of the A ring, which is mainly responsible for the antimicrobial activity of TCs [12], was performed in a one-step reaction, as shown in Figure 1a. As previously reported, DMC was treated with zinc in acetic acid and water for 2 h at room temperature [23,24,25]. After purification with preparative HPLC, the NMR data obtained from the compound revealed the complete structure of the molecule. Surprisingly, in addition to the efficient removal of the DMA group, the reaction also caused the reduction of the hydroxyl group at position 12a. It is important to note that these chemical modifications did not affect the region that was previously attributed to binding aggregated α-Syn species [26]. The double reduction of DMC led to a molecule that we named DDMC.

DDMC and DMC were evaluated for their antimicrobial activities against both Gram-negative (*Pseudomonas aeruginosa* PA01 and *Escherichia coli* ATCC 25922) and Gram-positive bacteria (*Staphylococcus aureus* ATCC 25923). DMC exhibited the most pronounced antibacterial activity against these bacteria, with minimum inhibitory concentrations (MICs) of 6.25 μM, 3.125 μM, and 0.4 μM, respectively (Figure 1b). As expected, with the removal of the DMA group, DDMC presented only residual antibiotic activities against Gram-negative bacteria while showing a 30-fold reduction in antibiotic activity against Gram-positive bacteria (MIC of 12.5 µM), with respect to that obtained for other TCs [27] (Figure 1b).

DMC is clinically used to treat susceptible bacterial infections and can be systemically administrated due to its low toxicity in human cells, even neurons [28]. In order to uncover if this was also the case for DDMC, we first evaluated its impact on cell viability by measuring MTT turnover [17] in SH-SY5Y cell cultures, using similar concentrations as those used by Braun et al. for DMC [13]. The results showed no statistical differences between both compounds, tested at 10 and 50 µM, with respect to the control conditions (Figure 1c). This finding allowed us to use both concentrations to study the neuroprotective properties of DDMC.

Given that dysfunctions in the lysosomal pathway have been proved to be a key player in neurodegenerative diseases [29,30], we used LysoTracker^TM^ Deep Red to test if DDMC could affect these organelles; this dye accumulates specifically in acidic vesicles and is widely used for determining the number and location of lysosomes [31]. As shown in Figure 1d, the overnight treatment of SH-SY5Y-α-Syn-tRFP cells with 10 μM of DDMC did not induce any changes in the number or intensity of LysoTracker^TM^-positive puncta. We also assayed the pan-lysosomal marker lysosomal-associated membrane protein 1 (LAMP1) [31] using immunofluorescence. As has been shown previously by others, the untreated SH-SY5Y- α-Syn-tRFP cells displayed abundant LAMP-1-positive vesicles (Figure 1e) [32]. Again, after an overnight treatment with DDMC, no significant differences were observed compared to the untreated samples, suggesting that this compound does not affect the number or distribution of lysosomes, which represent a crucial clearance mechanism for the protein aggregates that are thought to trigger neurodegenerative diseases [29,30].

### 3.2. Comparative Effect of DMC and DDMC on the Inhibition of α-Syn Aggregation

The ability of DMC to interfere with α-Syn amyloid-like aggregation has been previously reported [13]. However, its antibiotic activity would hinder its repositioning as a neuroprotectant. To study if DDMC retained the antiaggregant property shown by its precursor DMC, we monitored the ability of both TCs to inhibit α-Syn self-assembly. For this, 70 μM of fresh α-Syn protein was incubated in the absence or presence of 10 and 50 μM of each TC at the same concentration previously tested by Braun et al. [13]. The cross-β structure, which is the hallmark of amyloid aggregation [33], was monitored by thioflavin T (ThT) fluorescence emission at 482 nm (λ_exc_ 450 nm) [20].

In the presence of 50 μM DMC, the ThT fluorescence intensity—as expected—decreased by around 75% (*p* < 0.0001) compared to the control, while the effect at 10 μM was significantly less pronounced (26%, *p* = 0.0148) (Figure 2a). Surprisingly, DDMC was more effective than its precursor; at 50 μM, the ThT signal was almost completely abolished (*p* < 0.0001); moreover, at 10 μM, DDMC remained capable of inhibiting α-Syn cross-β formation, reaching a significant ThT signal reduction of 65% (*p* < 0.0001) (Figure 2a); these data strongly suggest that the chemical modifications introduced to DMC improved the ability of DDMC to interfere with α-Syn amyloid-like aggregation.

The binding mode(s) of molecular dyes to amyloid fibrils is not fully understood. However, there is some consensus suggesting that ThT binds in channels running parallel to the long axis of the fibril [34], while Congo red (CR), binds in grooves formed along the β-structure [35]. Therefore, both assays are complementary and, when used together provide deeper information on amyloid formation. Consequently, we complemented our results by studying the bathochromic shift of CR absorption spectrum [19,36]. As shown in Figure 2b, 10 µM DDMC (i.e, the lowest concentration tested) diminished the CR signal by 43% (*p* < 0.0001) with respect to the control, while DMC showed no significant spectral difference (*p* = 0.0735). Although the red shift in CR lacks sensitivity compared to ThT at low amyloid fibril concentrations [37], the results obtained in Figure 2b are in agreement with the ThT fluorescence data depicted in Figure 2a.

To confirm these results, we resorted to TEM to visualize the effects of DMC and DDMC on α-Syn aggregation. Figure 2c shows that DMC and DDMC both interfered with fibril formation; however, the reduction in fibrillar species formed in the samples incubated with DDMC compared to those treated with DMC was consistent with the decrease in cross-β structures depicted in Figure 2a. From these results, we concluded that DDMC was more potent than DMC at inhibiting α-Syn amyloid-like aggregation.

### 3.3. Effect of DDMC on α-Syn Preformed Fibril Seeding Activity

According to the nucleation/polymerization model, preformed α-Syn fibrillar species (α-Syn_PFF_) have the ability to induce and accelerate the aggregation of the soluble monomeric protein and could be considered as seeds in this reaction [38]. To assess the ability of DDMC to inhibit the seeding competence of α-Syn_PFF_, we compared the aggregation kinetics of monomeric α-Syn (α-Syn_m_) to α-Syn_m_ seeded with α-Syn_PFF_ in the absence (α-Syn_m_+α-Syn_PFF_) or presence of 10 µM DDMC (α-Syn_m_+α-Syn_PFF_+DDMC). As shown in Figure 3a, the α-Syn amyloid-like fibril formation kinetics displayed a typical sigmoidal curve. In fact, we observed a 15 h lag phase followed by exponential growth that finally reached a plateau at 48 h, as was previously described by González-Lizárraga and colleagues [10]. The addition of α-Syn_PFF_ to the α-Syn_m_ solution accelerated the aggregation process, as revealed by the hyperbolic kinetics that was fit to a hyperbolic function (Figure 3a). Surprisingly, the presence of DDMC at the beginning of the aggregation process caused a significant decrease in ThT fluorescent intensity at all time points evaluated (Figure 3a), and a 6 h lag phase was observed (Figure 3b).

### 3.4. Effect of DDMC on α-Syn_PFF_-Induced Seeding of Cytoplasmic α-Syn

Despite the fact that α-Syn_PFF_ formed from soluble α-Syn_m_ differ at the atomic level from those recovered from the brains of PD patients [39], these species can seed the aggregation of endogenous α-Syn in different models, including cultured neuronal cells, primary cultured neurons, and animal brains [40,41,42]. Considering the ability of DDMC to inhibit the seeding competence of α-Syn_PFF_, we complemented our biophysical studies with a dopaminergic cell model. For this, we used an SH-SY5Y transgenic cell line that constitutively overexpresses α-Syn fused to tRFP (Innoprot #P30707-02). In the absence of α-Syn_PFF_, the fusion protein was expressed in most cells with a diffuse and faint cytoplasmic pattern (Figure 4a). On the contrary, the addition of exogenous α-Syn_PFF_ induced the formation of intense α-Syn-tRFP puncta, which strongly resembled α-Syn aggregates (Figure 4b) [43,44]; however, when α-Syn_PFF_ was added together with 10 µM DDMC, these puncta were diminished (Figure 4c). The quantification of these tRFP-positive puncta, expressed as granularity (described in Materials and Methods), showed a significant increase from 0.46 to 0.63 after α-Syn_PFF_ addition; in cells treated with α-Syn_PFF_+DDMC, the values were significantly reduced to the levels shown in control cells. These results suggested that the seeding competence of α-Syn_PFF_ was efficiently inhibited by DDMC.

### 3.5. DDMC Inhibits the Inflammogenic Effects of α-Syn Amyloid Fibrils

The inflammatory potential of α-Syn depends on the state of protein aggregation. As was previously shown by our group, aggregated species such as α-Syn_PFF_ have strong inflammogenic properties in microglial cells [45]. These aggregates trigger the release of inflammatory factors, such as tumor necrosis factor α (TNFα) [45] and glutamate (Glut) [11]. In order to evaluate the ability of DDMC to interfere with the release of proinflammatory factors, we treated primary microglial cells with α-Syn_PFF_. After 24 h of treatment, tenfold and eightfold increases in TNFα and Glut release, respectively, were observed (Figure 5a,b). In the presence of DDMC, (α-S_PFF_+DDMC) a significant reduction the release of TNFα and Glut compared to the absence of the novel TC was observed. In addition, we also tested the inflammogenic activity of the remaining fibrils formed in the presence of DDMC, shown in Figure 2c (α-Syn_PFF_:DDMC), by measuring TNFα and Glut release in microglia cells. In order to exclude any effects due to the DDMC itself in the α-Syn_PFF_:DDMC samples, the TC was removed by dialysis before treatment (see Materials and Methods). As expected, α-Syn_PFF_ strongly stimulated the release of TNFα (Figure 5a) and Glut (Figure 5b) after 24 h compared to the control condition (*p* < 0.0001). However, when the cultures were exposed to α-Syn_PFF_:DDMC (fibrils formed in the presence of DDMC but with no residual DDMC), the release of TNFα and Glut was significantly decreased compared to α-Syn_PFF_, reaching levels comparable to the control condition.

## 4. Discussion

To this day, there is no approved disease-modifying treatment for PD, even less for one that could pharmacologically target the formation of α-Syn toxic aggregates or the inflammatory effect of the aggregated protein. This highlights an unmet need in the development of novel strategies for PD treatment. High-throughput screening systems (HTS) based on the inhibition of α-Syn toxic species provide starting chemical matters for drug discovery, and different selection platforms are constantly providing small molecules as hit compounds for PD [13,22,46]. These molecules must be studied in the subsequent steps of the drug design pipelines in order to obtain leaders for preclinical and clinical trials. Recently, Braun et al. (2021) identified DMC and the bi-benzimidazole Ro 90-7501 as compounds that prevented α-Syn aggregation and protected SH-SY5Y cells from αSyn-induced death [13]. Of these compounds, DMC was shown to be the most potent, as being effective at a nanomolar range. Besides, DMC could also inhibit the pathological phosphorylation of α-Syn Ser residue 129 phosphorylation in a murine primary neuron assay [13]. Therefore, this well-known first-generation TC emerged as an exciting candidate for the development of an efficient therapy for PD.

In the present work, we selected DMC as a starting material for chemical modifications in the hit-to-lead pipeline. Extensive knowledge regarding the functional properties of TC ring substituents allowed for the chemical modifications to be carried out in a rational way. So, the efficient removal of the DMA group in a one-step reaction, as expected, diminished the antibiotic activity of the molecule. The resulting non-antibiotic DDMC possesses the advantage of overcoming the potential risk of inducing antibiotic resistance in long-term administration. Importantly, this novel TC did not alter SH-SY5Y cell viability at a concentration of 50 μM, demonstrating that the new molecule was not toxic in the assays and conditions tested, a property that is not necessarily shared by all molecules uncovered by HTS assay.

Upon entering the cell, aggregated α-Syn is retained in lysosomes for at least a week [47], and may be proteolytically processed and/or cause lysosomal impairment [48]. However, a small amount of this misfolded α-Syn finds its way through the cytoplasm [49]. Lysosomal dysfunction has been strongly associated to aggregate-prone protein deposition, worsening the degradation of abnormal protein aggregates [50]. An increasing number of small molecules have demonstrated the ability to impair lysosomes [50]. Jiang et al. demonstrated that a TC pretreatment could indirectly inhibit autophagy activation in ischemic brain tissues [51]. On the contrary, Lim and Hyun demonstrated that minocycline treatment induced autophagy/lysosome-related gene expression in several tissues and reduced the accumulation of poly-ubiquitinated proteins [52]. The results presented here show that the novel TC, DDMC, did not affect the number or cellular localization of lysosomes in a cellular model expressing α-Syn-tRFP. These data acquire relevance when considering that lysosomal integrity is an essential point to consider when searching for new small molecules interferering with α-Syn-induced pathological events.

It is important to notice that the chemical modifications made to DMC were performed in a one-step reaction, which represents an advantage when considering a scaling-up in the synthesis of DDMC for testing in animal models and clinical trials. Moreover, the removal of the DMA group together with the reduction of the hydroxyl in position 12a of ring A increased the efficiency of this molecule for inhibiting amyloid-like aggregation of α-Syn (Figure 2). Accordingly, the ThT and Congo red spectroscopy analysis showed that DDMC strongly inhibited the aggregation process of α-Syn at concentrations as low as 10 µM. In order to achieve that level of inhibition, 50 µM of the parental molecule DMC was required. According to our previous results, similar inhibition with doxycycline or CMT-3 was obtained at concentrations of 100 µM [11]. This improvement could be useful in future clinical applications.

The data presented herein for DDMC also support our previously proposed amyloid-binding structural motif [10,11]. The fact that specific small molecules had been shown to target the amyloid-like aggregation of different proteins as diverse as PrP, tau, α-synuclein, and Aβ [53] led us to look for a common structural fingerprint that mediates this interaction. Using computational techniques, we uncovered a motif constituted by a proton donor/acceptor pair arrangement attached to a closed ring, arranged in an almost planar configuration at an average distance of 2.6 Å; this arrangement was structurally complementary to the protein backbone in the beta structure [54,55], and we proposed this configuration as responsible for interfering with cross-β formation. In the rational modification of DMC, we preserved this signature (Figure 1, dashed line box). The ability of DDMC to interfere with the α-Syn amyloid-like aggregation supports the relevance of this signature in inhibiting the formation of toxic species.

An ideal strategy to prevent PD progression would target not only the formation but also the spread of α-Syn amyloid-like aggregates between close neurons. Growing experimental evidence suggests that the progression of α-Syn pathology is mediated by aggregate-driven or prion-like models of propagation [56,57]. To study this process in cell cultures, Tanik et al. developed a model where α-Syn_PFF_ act as templates for endogenously expressed α-Syn, which leads to the subsequent formation of insoluble aggregates resembling LBs and Lewy neurites [43,58]. A compound capable of interfering with this injurious process, would be useful not only to avoid *de novo* aggregation of native α-Syn but also to inhibit the prion nucleation effect of previously aggregated species. Interestingly, as shown in Figure 3, the seeding ability of pre-aggregated α-Syn was significantly reduced in the presence of DDMC, both in biophysical assays and cell culture model. Indeed, our data demonstrate that 10 µM of the novel TC was highly efficient in reducing the formation of α-Syn_PFF_-induced cytoplasmic inclusions induced by α-Syn_PFF_ through inhibition of the seeding process. This effect could also be demonstrated in a cell-free model. In our system, α-Syn aggregation followed a typical sigmoidal curve, with a latency time of 16 hs followed by an exponential phase that reached a stationary state at around 44 hs (Figure 3a), as previously described [10]. In the presence of α-Syn_PFF_, the fast aggregation of the monomeric protein abolished the lag phase. However, in the presence of 10 μM DDMC, α-Syn_PFF_ were inefficient at nucleating new monomers, and the lag phase was restored, reaching 8 h. This suggests that the novel TC prevents the seeding reaction, elicited by α-Syn_PFF_ supporting the data from the cell-based model. In fact, the seeding ability of α-Syn_PFF_ exogenously added to SH-SY5Y-tRFP cells was significantly reduced in the presence of DDMC. Considering that the precise molecular mechanisms underlying the spread and accumulation of α-Syn remain obscure, this data point to the relevance of small molecules with a conserved “amyloid binding motif” for preventing the spread of the pathology and strengthening the position of DDMC as a drug candidate to target not only the formation but also the seeding ability of toxic aggregate α-Syn species.

The well-known ability of some TCs to inhibit the release of proinflammatory factors was also conserved in DDMC, as depicted in Figure 4. Indeed, DDMC was capable of inhibiting TNF-α and Glut release induced by α-Syn_PFF_ in microglial cells_._ Another interesting aspect of our results is the fact that the inflammogenic properties of the remanent α-Syn fibrils formed in the presence of the novel TC were also strongly reduced. Therefore, the presence of less inflammogenic α-Syn species could contribute synergistically to the neuroprotective process.

In summary, being PD a multifactorial disorder where protein misfolding yields toxic α-Syn species that spread the pathology in neuronal cells and promote neuroinflammation, one may assume that DDMC may be a multi-target drug with potential therapeutic utility for the treatment of this disorder. The absence of antibiotic activity of DDMC may confer a particular advantage over other neuroprotective TC derivatives leading us to propose DDMC as a promising drug to be studied further in the drug discovery pipeline.

## Figures and Tables

**Figure 1 cells-11-02759-f001:**
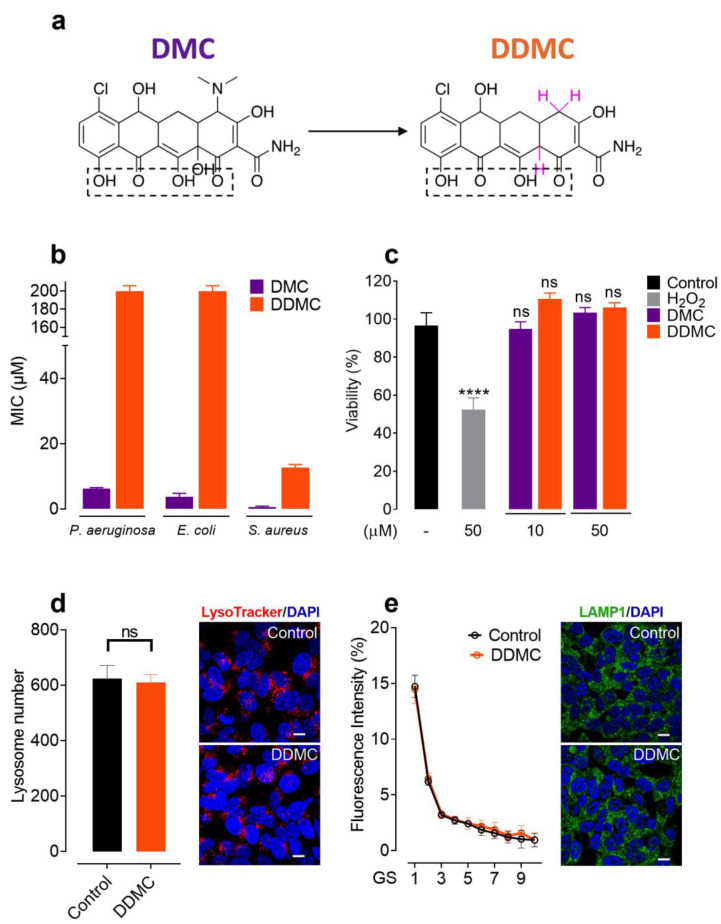
The novel TC DDMC is innocuous in bacteria and human cells. (**a**) Synthesis of DDMC via the treatment of DMC with zinc in acetic acid and water for 2 h at room temperature. The hydrogen atoms incorporated due to the reduction step are shown in pink. The dashed-line box points to the structural signature related to binding with amyloid-like aggregates. (**b**) Antibiotic activities of DMC and DDMC against *P. aeruginosa* (PAO1), *E. coli* (ATCC 25922), and *S. aureus* (ATCC 25923). MICs are shown in µM. (**c**) MTT cell viability assay in SH-SY5Y cells after 24 h treatment with DMC or DDMC (10 or 50 µM final concentrations) and hydrogen peroxide (H_2_O_2_) as a positive control (50 µM). All experiments were performed in sextuplicate, and relative cell viability was expressed as a percentage relative to the untreated control cells. Statistical analysis was performed as described in Materials and Methods, and significant differences are indicated in the figures as follows: **** *p* < 0.001 vs. control; ns (not significant)—*p* > 0.05 vs. control. (**d**) The effect of 10 µM DDMC on lysosome numbers was assessed using LysoTracker^TM^ Deep Red staining in SH-SY5Y cells and confocal microscopy. (**e**) The number of LAMP1-positive vesicles was estimated using immunostaining and captured with a confocal microscope. Lysosome quantification for (**c**) and (**d**) was performed with the granularity module in CellProfiler 4.2.1 [21].

**Figure 2 cells-11-02759-f002:**
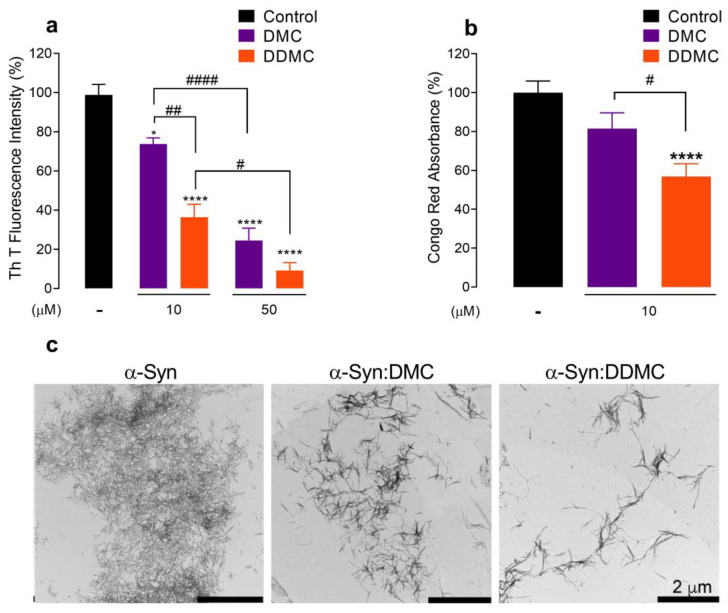
Comparative effect of DMC and DDMC on the inhibition of α-Syn aggregation. (**a**) Effect of DMC and DDMC on α-Syn aggregation as measured by the fluorescence emission intensity of 25 μM ThT in a solution containing 70 μM α-Syn incubated in the presence of 10 or 50 μM DMC or DDMC, after 120 h of incubation. Statistical significance is indicated as: * *p* = 0.0148; **** *p* < 0.0001 vs. Control (-); # *p* = 0.0152 DDMC 10 μM vs. DDMC 50 μM, ## *p* = 0.0025 DMC 10 μM vs. DDMC 10 μM, #### *p* < 0.0001 DMC 10 μM vs. DMC 50 μM. (**b**) Absorbance of 20 µM Congo red in a solution containing 70 μM α-Syn incubated in the presence or absence of 10 µM DMC or DDMC. Statistical significance is indicated as: **** *p* < 0.0001 vs. Control (-); # *p* = 0.025 DMC 10 μM vs. DDMC 10 μM. Values represent the mean ± SEM (*n* = 5), and statistical analyses were performed as described in Materials and Methods. (**c**) Transmission electron microscopy (TEM) of α-Syn samples incubated for 120 h in the presence of 10 μM DMC or DDMC.

**Figure 3 cells-11-02759-f003:**
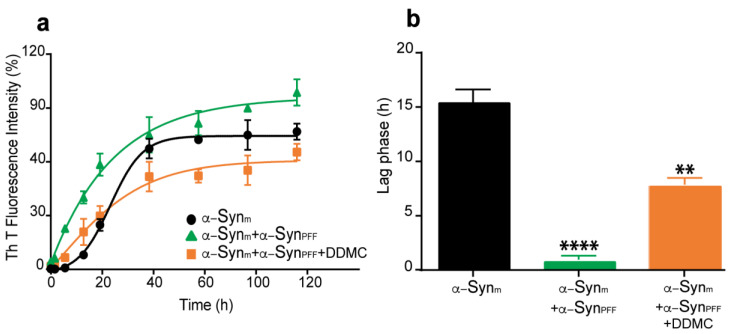
Effect of DDMC on α-Syn seeding induced by externally added α-Syn_PFF_. (**a**) The fluorescence emission intensity of 25 μM ThT as a function of time (0-120 hr) in a solution containing 70 μM monomeric α-Syn (α-Syn_m_) alone (black circles) or seeded with α-Syn_PFF_ (α-Syn_m_+α-Syn_PFF_) in the absence (green triangles) or presence of DDMC (α-Syn_m_+α-Syn_PFF_+DDMC) (orange squares) after 120 h of incubation. For each time point, data values represent the mean + SEM. (**b**) Lag phase quantification based on the kinetics of aggregation of the seeding assay as described in Materials and Methods. Statistical significance is indicated as: **** *p* < 0.01; ** *p* < 0.01 vs. α-Syn_m_.

**Figure 4 cells-11-02759-f004:**
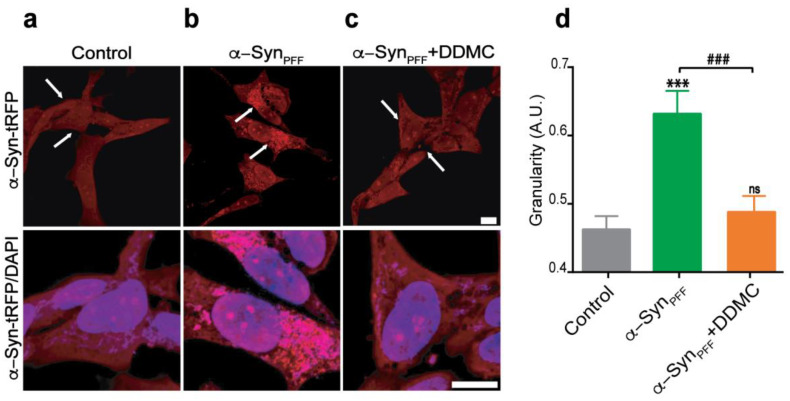
Effect of DDMC on α-Syn_PFF_-induced seeding of cytoplasmic α-Syn. (**a**–**c**) SH-SY5Y transgenic cell line (Innoprot #P30707-02) expressing α-Syn-tRFP treated with α-Syn_PFF_ or α-Syn_PFF_+DDMC (10 µM). The lower panel corresponds to an enlargement of the region indicated with the white arrows together with DAPI counterstaining. (**d**) Granularity was quantified using CellProfiler 4.2.1. Statistical significance is indicated as: *** *p* < 0.001 vs. Control; ### *p* < 0.01 vs. α-Syn_PFF_; ns not significant vs. Control.

**Figure 5 cells-11-02759-f005:**
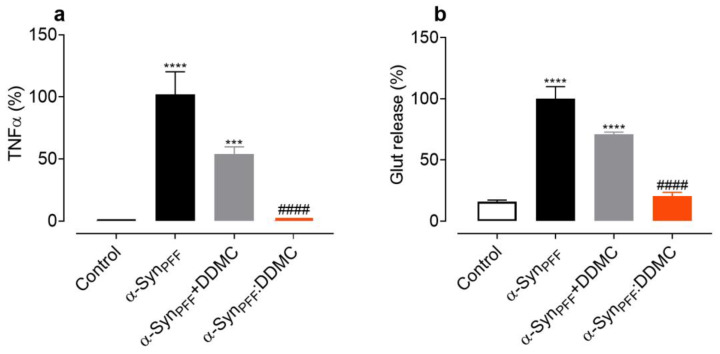
DDMC mitigates the inflammogenic properties of α-Syn amyloid-like fibrils. (**a**) TNFα and (**b**) Glut release from microglial cells undergoing (i) no treatment (control) or an exposure to (ii) α-Syn_PFF_; (iii) α-Syn_PFF_ together with 0.7 µM DDMC (α-Syn_PFF_+DDMC) or to iv) α-SynPFF formed in the presence of 10 µM DDMC (α-SynPFF:DDMC) with subsequent dialysis. Bars represent the mean ± S.E.M (*n* = 8). Statistical significance is indicated as: *** *p* < 0.001; **** *p* < 0.001 vs. α-Syn_PFF_; #### *p* < 0.0001 vs. control; ns—*p* > 0.05. Note that 0.7 µM DDMC corresponds to the residual concentration of DDMC theoretically remaining in α-Syn_PFF_+DDMC samples before dialysis.

## Data Availability

The datasets generated during the current study are available from the corresponding author upon reasonable request.

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
