# Peer review of "Neuroprotective Effects of a Novel Demeclocycline Derivative Lacking Antibiotic Activity: From a Hit to a Promising Lead Compound"

_cells, 2022, doi:10.3390/cells11172759_

Round 1

Reviewer 1 Report

The manuscript entitled "Neuroprotective effects of a novel demeclocycline derivative lacking antibiotic activity: from a hit to a promising lead compound".

In this study, the authors have suggested that demeclocycline derivative with residual antibiotic activity improved neuroprotective effects. I have reviewed the manuscript and have given my comments in the following lines.

Major comments

1. The figure data is unclear for understanding the author's remarks. Is it possible to upgrade to make Figure 1d, e, and Figure 4a, b, c better quality? How about increasing the dpi of this figure?

2. In figure 1d, e the authors' argument is weak to defend their data on the lysosomal pathway. The author should supplement your argument by adding more data. And the authors mismarked the data about LAMP1 in the results part.

3. In the results part, it is weak to defend the author’s hypothesis that DDMC inhibits the α-Syn aggregation. As the author knows, the authors did not show the data about the phosphorylation of α-Syn. The thioflavin T fluorescence assay is not enough to remark about α-Syn aggregation.

4. Also, the authors suggested that the DDMC inhibited α-SynPFF. It is weak to defend the author’s hypothesis. Please back up this data.

5. How the DDMC dose was selected on the basis of in vitro studies? Are there previous studies available regarding the dose of DDMC? Please clarify this and mention here for future studies of other scientific communities.

6. The authors have shown that DDMC inhibits the inflammogenic effects of α-SynPFF. It is weak to defend the author’s hypothesis. In my suggestion; First, it would be better to add the data of protein level or mRNA level using western blots or qPCR, respectively. Second, show the morphology of microglia using IHC or IF.

7. There are a lot of overlapping remarks in the Discussion part. The authors should revise the Discussion part to make it concise. It has lots of content in this part. There are overclaimed summary statements as the authors have provided weak data on the link between DDMC and α-Syn toxic aggregation and neuroinflammation.

Minor comments

1. The authors used 'ex vitro’ word that I am wondering about the exact meaning.

2. The authors have shown the viability of human neuroblastoma SH-SY5Y cells under DMC and DDMC treatment using MTT assay. The authors would better indicate the remark ‘n.s.’ meaning not significant.

3. In figure 1e, please indicate what the authors detect marker using immunofluorescence assay.

4. The authors did not kindly explain the meaning of TC.

5. How many did the same experiments? The authors did not remark on the information in figure 5 data.

6. In the Introduction part, the authors should revise and edit to make it concise and compact. Consequently, this introduction part needs to be revised obviously so that the reader can understand the authors' meaning.

Typo

1. In the results part, the authors should change from the figure 1d word to figure 1e in the folloing sentence. As has been shown previously by others, the untreated SH-SY5Y- α-Syn-tRFP cells displayed abundant LAMP-1-positive vesicles (Figure 1d).

Final comments

The paper may be accepted if the author improves it to a better scientific level.

Best of luck to the authors

Reviewer 2 Report

This paper describes a new tetracycline that is devoid of antimicrobial capacity, thus is suitable for long term treatment as it would not confer antibiotic resistance. The  study is sound and the results are very promising.

Minor:

1)The quality of photomicrographs on figure 1 could be improved. 

2) studies on doxyclicline effects on other neurodegenerative disorders such as HD should be mentioned (Paldino eta l, 2020)

Round 2

Reviewer 1 Report

The paper may be accepted if the author improves it to a better scientific level.